# Reward Design in Cooperative Multi-agent Reinforcement Learning for Packet Routing

## Abstract

In cooperative multi-agent reinforcement learning (MARL), how to design a suitable reward signal to accelerate learning and stabilize convergence is a critical problem. The global reward signal assigns the same global reward to all agents without distinguishing their contributions, while the local reward signal provides different local rewards to each agent based solely on individual behavior. Both of the two reward assignment approaches have some shortcomings: the former might encourage lazy agents, while the latter might produce selfish agents.

In this paper, we study reward design problem in cooperative MARL based on packet routing environments. Firstly, we show that the above two reward signals are prone to produce suboptimal policies. Then, inspired by some observations and considerations, we design some mixed reward signals, which are off-the-shelf to learn better policies. Finally, we turn the mixed reward signals into the adaptive counterparts, which achieve best results in our experiments. Other reward signals are also discussed in this paper. As reward design is a very fundamental problem in RL and especially in MARL, we hope that MARL researchers can rethink the rewards used in their systems.

## 1 Introduction

In reinforcement learning (RL), the goal of the agent is formalized in terms of a special signal, i.e., reward, coming from the environment. The agent tries to maximize the total amount of reward it receives in the long run. Formally, we express this idea as the *Reward Hypothesis* (Sutton & Barto, 1998): the goal of RL agent can be exactly described as the maximization of the expected value of the cumulative sum of a received scalar reward signal.

It is thus critical that the rewards truly indicate what we want to accomplish. One reward design principle is that the rewards must reflect *what* the goal is, instead of *how* to achieve the goal [1]. For example, in AlphaGo (Silver et al., 2016), the agent is only rewarded for actually winning. If we also reward the agent for achieving subgoals such as taking its opponent's pieces, the agent might find a way to achieve them even at the cost of losing the game. A similar example of faulty reward function is provided by Russell & Norvig (2010): if we reward the action of cleaning up dirt, the optimal policy causes the robot to repeatedly dump and clean up the same dirt. In fact, the *how reward* encodes human experience, which is heuristic in some extent. Based on the heuristic *how reward*, it is really easy to deviate from the ultimate goal.

However, as van Seijen et al. (2017) point out, the exact *what reward* that encodes the performance objective might be awful to use as a training objective. It will result in slow and unstable learning occasionally. At the same time, a training objective that differs from the performance objective can still do well with respect to it. For example, the Intrinsically Motivated Reinforcement Learning (IMRL) (Chentanez et al., 2005; Shaker, 2016) combines a domain-specific *intrinsic reward* with the reward coming from the environment to improve learning especially in sparse-reward domains.

Although reward design problem in single-agent RL is relatively tractable, it becomes more thorny in multi-agent reinforcement learning (MARL), as MARL is naturally more complex than single-agent RL. As we know, the global reward and local reward have long been proved to be defective: the former might encourage lazy agents, while the latter might produce selfish agents.

---

[1] In this paper, we call these two kinds of rewards *what rewards* and *how rewards*, respectively.

Inspired by the success of *intrinsic reward* in single-agent RL, we hypothesize that similar methods may be useful in MARL too. Naturally, in this paper, we ask and try to answer a question:

> Can we formulate some special rewards (such as *intrinsic reward*) based on the meta *what rewards* to accelerate learning and stabilize convergence of MARL systems?

Specifically, in this paper, we propose several new MARL environments modified from the well known Packet Routing Domain. In those environments, the goal is to figure out some good flow splitting policies for all routers (i.e., agents) to minimize the maximum link utilization ratio in the whole network. We set the meta reward signals as $1 - \max(U_l)$. We argue that the meta reward signals are some kinds of *what rewards* because they tell the agents that we want to minimize $\max(U_l)$, i.e., minimize the maximum of all link utilization ratios. For detailed discussions, we refer the readers to the proposed environments and rewards in Section 3 and 4.

Based on those environments and the meta *what rewards*, we can focus on our reward design research purpose. Specifically, we firstly show that both of the widely adopted global and local reward signals are prone to produce suboptimal policies. Then, inspired by some observations and considerations, we design some mixed reward signals, which are off-the-shelf to learn better policies. Finally, we turn the mixed reward signals into the adaptive counterparts, which achieve best results in our experiments. Besides, we also discuss other reward signals in this paper.

In summary, our contributions are two-fold. (1) We propose some new MARL environments to advance the study of MARL. As many applications in the real world can be modeled using similar methods, we expect that other fields can also benefit from this work. (2) We propose and evaluate several reward signals in these MARL environments. Our studies generalize the following thesis (Chentanez et al., 2005; van Seijen et al., 2017) in single-agent RL to MARL: agents can learn better policies even when the training objective differs from the performance objective. This remind us to be careful to design the rewards, as they are really important to guide the agent behavior.

The rest of this paper is organized as follows. Section 2 introduces background briefly, followed by the proposed environments and rewards in Section 3 and 4, respectively. We then present the experiments and discussions in Section 5 and 6, respectively. Section 7 concludes this work.

## 2 BACKGROUND

As reward is the foundation of RL, there are many related studies. We only introduce the most relevant fields of this work. Topics such as Inverse RL (Ng & Russell, 2000) and Average Reward RL (Mahadevan, 1996) are not included.

### 2.1 REWARD DESIGN

The only way to talk with your agents might be the reward, as expressed by the well known *Reward Hypothesis* (Sutton & Barto, 1998).

When considering reward design for single objective RL problem, we should always be aware of whether the designed reward is a kind of *what reward* rather than *how reward*. For multiple objectives RL problem, researchers have to design sub-rewards for different objectives, and the final reward is a weighted sum of those sub-rewards. Unfortunately, the weights have to be adjusted manually even in recent studies (Hausknecht & Stone, 2015; Li et al., 2016; Diddigi et al., 2017).

For single-agent RL, there are many remarkable reward design studies. The most relevant field may be the IMRL. A recent Deep RL work based on IMRL is VIME (Houthooft et al., 2016). It uses a surprise reward as the *intrinsic reward* to balance exploitation and exploration, and achieves better performance than heuristic exploration methods. Another successful model is Categorical DQN (Bellemare et al., 2017), which considers the long time run reward, or value, used in approximate RL. The authors model the value distribution rather than the traditional expected value. They obtain anecdotal evidence demonstrating the importance of the value distribution in approximate RL. Li et al. (2017) use the temporal logic (TL) quantitative semantics to translate TL formulas into real-valued rewards. They propose a temporal logic policy search (TLPS) method to solve specify tasks and verify its usefulness.

However, the reward design studies for MARL is so limited. To the best of our knowledge, the first (and may be the only) reward design study about Deep MARL is the well known Independent DQN (Tampuu et al., 2015). By setting different rewarding schemes of Pong, the authors demonstrate how competitive and collaborative behaviors emerge. Although the rewards used are very elaborate, there are only two agents in Pong, which limits its generalization ability.

## 2.2 CREDIT ASSIGNMENT

Credit assignment usually has two meanings. In single-agent RL, it is the problem that how much credit should the current reward be apportioned to previous actions. In MARL, it is the problem that how much credit should the reward obtained at a team level be apportioned to individual agents. In fact, credit assignment is a subclass of reward design, and the two fields can often be seen as the same in MARL. As far as we know, credit assignment approaches in MARL can be divided into two categories: traditional global/local reward and other rewards.

The global reward approach assigns the same global reward to all agents without distinguishing their contributions. On one hand, the lazy agents often receive higher rewards than what they really contribute to, which leaves lazy agents with no desire to optimize their policies. On the other hand, the diligent agents can only receive lower rewards even when they generate good actions, as the lazy agents often make the whole system worse. This makes the diligent agents confuse about what is the optimal policy. The local reward approach provides different rewards to each agent based solely on its individual behavior. It discourages laziness. However, agents do not have any rational incentive to help each other, and selfish agents often develop greedy behaviors.

Other reward signals need more complex computation or depend on non-universal assumptions. For example, Mataric (1994) uses the social reward at the cost of observing and imitating the behavior of other agents, while Agogino & Turner (2005); Tumer & Agogino (2007); Rădulescu et al. (2017); Foerster et al. (2017) use more computation to calculate the *difference reward* or *CLEAN reward* (counterfactual baseline). Chang et al. (2004b) assume that the observed reward signal is the sum of local reward and external reward that is estimated using a random Markov process, while Marbach et al. (2000); Sunehag et al. (2017); Cai et al. (2017) explicitly model the global reward as the sum of all local rewards.

## 3 THE PROPOSED ENVIRONMENTS

After having considered several other options, we finally choose the Packet Routing Domain as our experimental environments. The reasons are as follows.

- Firstly, they are classical MARL environments. There are many researchers studying packet routing problem (Boyan & Littman, 1994; Choi & Yeung, 1996; Subramanian et al., 1997; Wolpert et al., 1999; Chang et al., 2004a; Tillotson et al., 2004; Winstein & Balakrishnan, 2013; Dong et al., 2015; Rădulescu et al., 2017; Mao et al., 2017).

- Besides, many real-world applications can be modeled using similar methods, for example, the internet packet routing, electric power supply, natural gas transportation and traffic flow allocation. We expect that these fields can also benefit from this work.

- And most importantly, the *what reward* of these environments can be easily figured out, so we can focus on our reward design research purpose. Specifically, in these environments, the goals are high throughput and low link utilization ratio. In the real world, we also assume that the whole network capacity is bigger than the total flow demands [2], so the throughput is equal to the flow demands if we can find good packet routing policies. With this assumption, we set the meta reward signals as 1 - $\max(U_l)$. We argue that the meta rewards are some kinds of *what rewards* because they tell the agents that we want to minimize $\max(U_l)$, i.e., minimize the maximum of all $U_l$.

- Another attractive feature of those environments is that when we compute the global reward signal, the local reward signals of each agent can also be calculated without too much additional computation. See the detailed discussions in Section 4.

---

[2] Otherwise, we need upgrade the network devices except for finding good packet routing policies.

More concretely, we consider the environments as shown in Figure 1, 2 and 3. Currently, the Internet is made up of many ISP networks. In each ISP network, as shown in Figure 1, there are several edge routers. Two edge routers are combined as ingress-egress router pair (IE-pair). The $i$-th IE-pair has a input flow demand $F_i$ and $K$ available paths that can be used to deliver the flow from ingress-router to egress-router. Each path $P_i^k$ is made up of several links and each link can belong to several paths. The link $L_l$ has a flow transmission capacity $C_l$ and a link utilization ratio $U_l$. As we know, high link utilization ratio is bad for dealing with bursty traffic, so we want to find some good flow splitting policies jointly for all IE-pairs across their available paths to minimize the maximum link utilization ratio in the ISP network. More formally, the goal is the same as Kandula et al. (2005):

$$\min U_{l^*} \tag{1}$$

subject to the constraints:

$$l^* = \arg\max_l U_l \tag{2}$$

$$U_l = \sum_{F_i} \sum_{L_l \in P_i^k} \frac{F_i * y_i^k}{C_l} \tag{3}$$

$$1 = \sum_k y_i^k \tag{4}$$

$$0 \le y_i^k \tag{5}$$

where $F_i * y_i^k$ is the flow transmitted along path $P_i^k$ and $y_i^k$ is the corresponding ratio of $F_i$. Here, the policy $\pi(a|s)$ is approximated using DNN $f(y_i^k|s;\theta)$, and we need $y_i^k$ for the flow routing.

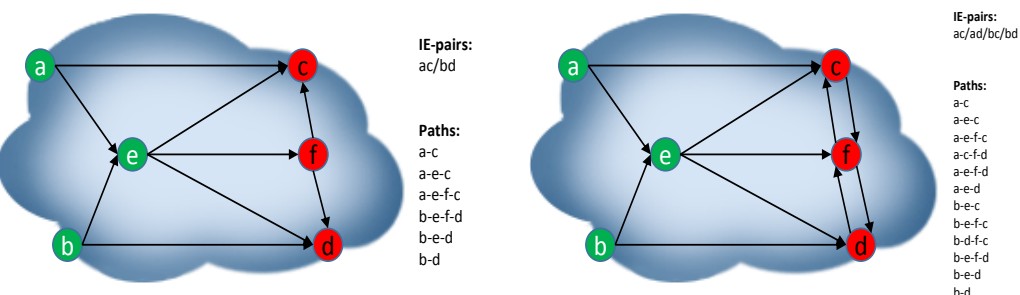

Figure 1: Simple Topology environment.    Figure 2: Moderate Topology environment.

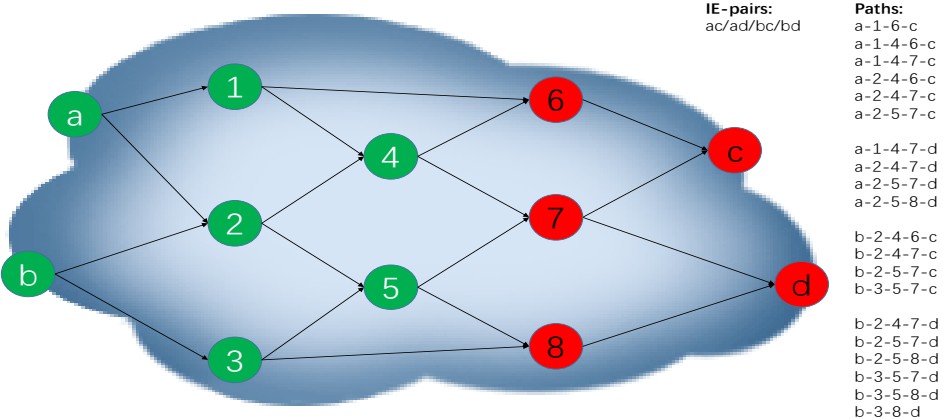

Figure 3: Complex Topology environment.

## 4 THE DESIGNED REWARDS

Before the definition of the designed rewards, we firstly define some link utilization ratio sets.

- *ALL* set of any router. It contains the link utilization ratios of ALL links in the network. For example, in Figure 1, $ALL = \{U_{ac}, U_{ae}, U_{bd}, U_{be}, U_{ec}, U_{ed}, U_{ef}, U_{fc}, U_{fd}\}$.

- *DIRECT* set of a given router. It only contains the link utilization ratios of the DIRECT links of this router. For example, in Figure 1, $DIRECT(e) = \{U_{ec}, U_{ed}, U_{ef}\}$.

- *BASIN* set of a given router. It only contains the link utilization ratios of the BASIN links of this router. And the BASIN links of a router are the links that this router can route flow on. For example, in Figure 1, $BASIN(e) = \{U_{ec}, U_{ed}, U_{ef}, U_{fc}, U_{fd}\}$.

Now, we are ready to define the designed reward signals as follows. A simplified illustration of these reward signals is shown in Figure 4.

- **Global Reward (gR).** The reward is calculated based on *ALL* set: $gR = 1 - max(ALL)$.

- **Direct Local Reward (dlR).** The reward is calculated only based on *DIRECT* set of a given router. For example, $dlR(e) = 1 - max(DIRECT(e))$.

- **Basin Local Reward (blR).** The reward is calculated only based on *BASIN* set of a given router. For example, $blR(e) = 1 - max(BASIN(e))$.

- **Direct Mixed Reward (dlgMixedR).** The reward is the sum of gR and dlR. For example, $dlgMixedR(e) = gR + dlR(e)$.

- **Basin Mixed Reward (blgMixedR).** The reward is the sum of gR and blR. For example, $blgMixedR(e) = gR + blR(e)$.

- **Direct Adaptive Reward (dlgAdaptR).** The reward is the sum of gR and w*dlR, where w is the weight of dlR. We decay w gradually at the training procedure. For example, $dlgAdaptR(e) = gR + w * dlR(e)$.

- **Basin Adaptive Reward (blgAdaptR).** The reward is the sum of gR and w*blR, where w is the weight of blR. We decay w gradually at the training procedure. For example, $blgAdaptR(e) = gR + w * blR(e)$.

From the above definitions, we can see that gR is *what reward*. It tells the agents that we want to minimize the maximum of all $U_l$. Besides, both dlR and blR can be seen as some kinds of *what rewards* from the local perspective of individual agent. Specially, both mixed rewards and adaptive rewards are simple combinations of those *what rewards*. Despite the simplicity of those designed rewards, we can focus on our research purpose: can those rewards accelerate learning and stabilize convergence of MARL systems? And as mentioned in previous sections, we can calculate these rewards at a low cost, because *DIRECT* set and *BASIN* set are subsets of *ALL* set.

To make our contributions more clearly, please note that traditional packet routing studies often use $gR$ and $dlR$, all other signal forms are firstly introduced in this paper as far as we know.

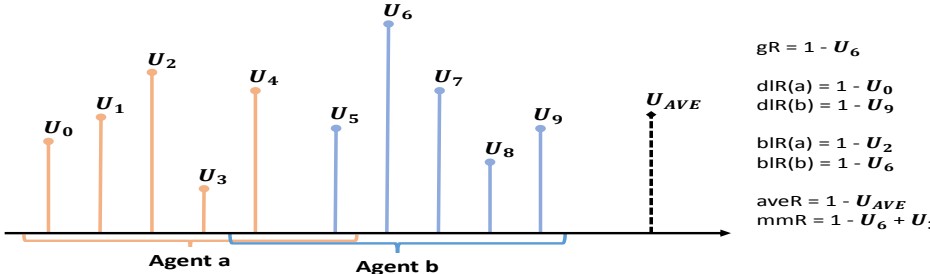

Figure 4: A simplified illustration of different reward signals.

## 5 THE EXPERIMENTS

In order to make the experimental environments consistent with the real-world systems, we highlight the following setting. More detailed information can be found in the Appendix Section A.1 and A.2.

- The routers are partially observable as they are located in different places. We use the recent proposed ACCNet (Mao et al., 2017) as the experimental method to handle this problem.
- The time delay of router, link and reward signal cannot be neglected. That is to say, actions have long term effect on the environments, which makes the task more challenging.
- Both synthetic flow and real flow trajectory from the American Abilene Network [3] are used to test the proposed rewards.

### 5.1 MACROSCOPIC RESULTS

For different rewards, we run 10 independent experiments on different topologies, and the averaged convergence rates (CR) and maximum link utilization ratios ($U_{l*}$) are shown in Table 1. From the perspective of topology, we can see that the convergence rates of any reward signal are decreasing gradually as the topology becoming more complex, which is reasonable. From the perspective of reward, we draw the following conclusions.

Table 1: The averaged CR and $U_{l*}$ using different rewards on different topologies.

| Reward Type | Simple Topology | | Moderate Topology | | Complex Topology | |
|---|---|---|---|---|---|---|
| | CR(%) | $U_{l*}$ | CR(%) | $U_{l*}$ | CR(%) | $U_{l*}$ |
| gR | 30 | 0.767 | 30 | 0.733 | 10 | 0.8 |
| dlR | 50 | 0.8 | 40 | 0.725 | 30 | 0.75 |
| blR | 50 | 0.74 | 50 | **0.62** | 20 | 0.7 |
| dlgMixedR | 60 | 0.767 | 50 | 0.64 | 50 | 0.72 |
| blgMixedR | 80 | 0.75 | **90** | 0.678 | 30 | 0.733 |
| dlgAdaptR | 70 | **0.7** | 70 | 0.671 | **60** | 0.733 |
| blgAdaptR | **90** | 0.729 | **90** | 0.725 | 50 | **0.65** |

Firstly, both $dlR$ and $blR$ are better than $gR$, which means that the widely used $gR$ in other MARL environments is not a good choice for the packet routing environments. The bad performance of $gR$ motivates us to discover other more effective reward signals.

Importantly, the proposed $blR$ seems to have similar capacity with $dlR$, but when we consider mixed reward signals and adaptive reward signals, the differences between them are obvious. For example, $blgMixedR$ and $blgAdaptR$ can achieve higher convergence rates than $dlgMixedR$ and $dlgAdaptR$ on Simple Topology and Moderate Topology, while $dlgMixedR$ and $dlgAdaptR$ has better performance on Complex Topology than $blgMixedR$ and $blgAdaptR$. In our opinion, $dlR$ has low $factoredness$ but high $learnability$, while $blR$ can better balance $factoredness$ and $learnability$ [4], which makes $dlR$ more suitable for symmetrical Complex Topology and $blR$ more suitable for asymmetrical Simple Topology and Moderate Topology. Anyways, the proposed $blR$ is very necessary for the packet routing environments.

Besides, on the whole, the adaptive reward signals are better than the mixed reward signals, and both of them are better than the meta rewards (we refer to $gR$, $dlR$ and $blR$). But please note that the mixed reward signals can achieve good results without any change of the experimental setting [5], while the adaptive reward signals have to adjust the replay buffer size and the weight decay rate.

Finally, we also notice that the proposed reward signals cannot effectively decrease $U_{l*}$ on Simple Topology. However, when we consider Moderate Topology and Complex Topology, the best reductions of $U_{l*}$ are bigger than 10%. The reason is that all rewards can approach the optimal policies

---

[3]https://en.wikipedia.org/wiki/Abilene_Network

[4]Agogino & Turner (2005) discuss $factoredness$ and $learnability$ more formally.

[5]That is why we use "off-the-shelf" to describe the mixed reward signals.

on Simple Topology, which leaves no space for the proposed rewards to further improve. But when the topology becomes more complex, the proposed rewards begin to show their abilities.

In short, our conclusions are: both global reward and local reward are prone to produce suboptimal policies; the mixed reward signals are off-the-shelf to learn better policies; and the adaptive reward signals can achieve best results at the cost of careful experimental setting.

Those conclusions also generalize the following thesis in single-agent RL to MARL: the training objective can differ from the performance objective, but still do well with respect to it. So we hope that MARL researchers can rethink the rewards used in their systems, especially when the agents cannot work as expected.

## 5.2 MICROSCOPIC ANALYSIS OF THE RESULTS

In this section, we highlight some observations and considerations that inspire us to design the above reward signals, based on Simple Topology in Figure 1.

**The gR and dlR.** We firstly try $gR$ without any thinking, but only get a 30% convergence rate, so we try the widely used $dlR$, and we find the performance is slightly better. However, we notice that the maximum or minimum link utilization ratio of most divergence experiments is $U_{fc}$ or $U_{fd}$, as shown in Figure 5. That is to say, the $dlR$ cannot catch information of $L_{fc}$ or $L_{fd}$, which is a little far from the nearest agent e (router f is not a ACCNet agent). We realise that the combination of $gR$ and $dlR$ might have the ability to catch both nearest link information and far away link information, so we propose the $dlgMixedR$.

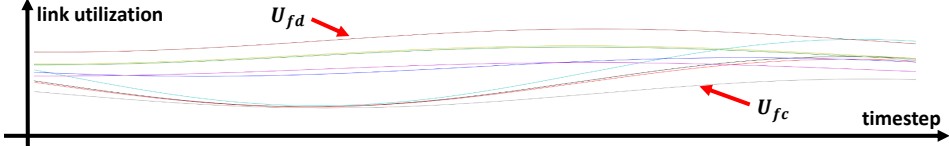

Figure 5: link utilization ratios of one divergence environment using $dlR$.

**The dlgMixedR.** As expected, the $dlgMixedR$ performs better than $gR$ or $dlR$ alone, not only for higher convergence rate but also lower $U_{l*}$. At this time, we further ask ourselves two questions. The first question is that can we simplify $dlgMixedR$ but still keep its ability? This inspires us to propose the $blR$ and subsequently $blgMixedR$. The second one is that can we gradually decay the weight of $dlR$ so that the received reward is progressively approaching $gR$, i.e., the performance objective of our environments? This inspires us to try the adaptive rewards.

**The blR and blgMixedR.** Although $blR$ is simpler than $dlgMixedR$, it can incorporate both nearest link information and far away link information adaptively. And as expected, $blR$ achieves similar convergence rates and maximum link utilization ratios as $dlgMixedR$. But what surprises us is that $blgMixedR$ can boost the convergence rate up to 80%. One hindsight is that $blR$ can better balance $factoredness$ and $learnability$.

**The dlgAdaptR and blgAdaptR.** Although the adaptive rewards get the best results, it is not until we actually implement them that we realize the difficulties. The results are sensitive to replay buffer size and weight decay rate, because the reward in replay buffer is slightly inconsistent with current reward even thought the $<s,a,s'>$ tuple is the same. Larger buffer size and higher weight decay rate usually mean greater inconsistency, while small buffer size and low weight decay rate result in slow learning and data inefficiency. So we suggest to use the off-the-shelf mixed reward signals.

**Other reward forms.** We also test min-max reward $mmR = 1 + min(U_l) - max(U_l)$ and average reward $aveR = 1 - average(U_l)$. The observation is that some links always have low link utilization ratios, which means that the agents have not learned to share flow on those links. So we try to encode those information into the reward using $mmR$ and $aveR$. Although those reward forms take some effect indeed, we do not consider them as general methods to discuss as they are not *what rewards*.

Finally, we give the link utilization ratios testing on real flow trajectory from the American Abilene Network, as shown in Figure 6. We see that all links have similar utilization ratios, and the trends of

the curves are consistent. Those results mean that all the links can properly share responsibility for the flow demand according to their respective capabilities.

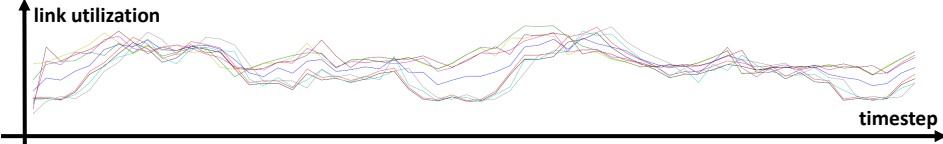

Figure 6: link utilization ratios testing on real Abilene Network flow using $blgMixedR$.

## 6 SOME DISCUSSIONS

**Why the convergence rate of Complex Topology is low?** In this paper, we only focus on designing special reward signals, rather than applying other sophisticated technologies, to solve the packet routing problem. In fact, the convergence rate can be improved to almost 100% for all topologies if we combine the proposed rewards with other methods. To make this paper easy to read, we do not introduce irrelevant methods.

**Can the proposed rewards generalize successfully in other environments?** In fact, not all environments can directly calculate the local reward or global reward, as Panait & Luke (2005) point out. In such environments, the proposed rewards might be only useful at high computation cost. However, the calculation of the rewards is not the research purpose of this paper. We argue that although the proposed rewards have limitations, they can be easily applied to many real-world applications such as internet packet routing and traffic flow allocation, as mentioned in Section 3.

**Can the designed rewards be seen as a kind of auxiliary task?** Yes, they are some auxiliary reward signals indeed. But please note that the auxiliary reward signals are different from the auxiliary task used in UNREAL (Jaderberg et al., 2016), where the auxiliary task is used for improving the feature extraction ability of neural networks, while our auxiliary reward signals directly guide the learned policies. In fact, the mixed rewards are similar with VIME (Houthooft et al., 2016) as analyzed in Section 2.1. And the adaptive rewards are similar with curriculum learning (Wu & Tian, 2016), as both of them train the agents progressively from easy to the final difficult environment.

## 7 CONCLUSION

In this paper, we study reward design problem in cooperative MARL based on packet routing environments. Firstly, we show that both of the widely adopted global and local reward signals are prone to produce suboptimal policies. Then, inspired by some observations and considerations, we design some mixed reward signals, which are off-the-shelf to learn better policies. Finally, we turn the mixed reward signals into the adaptive counterparts, which achieve best results in our experiments.

Our study generalizes the following thesis (Chentanez et al., 2005; van Seijen et al., 2017) in single-agent RL to MARL: the training objective that differs from the performance objective can still do well with respect to it. As reward design is a very fundamental problem in RL and especially in MARL, we hope that MARL researchers can rethink the rewards used in their systems.

For future work, we would like to use Evolutionary Algorithm (Fonseca & Fleming, 2014) to search the best weight of local reward, and verify whether the learned weight has the same decay property. We also expect to test the proposed reward signals in other application domains.

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

## A APPENDIX

### A.1 DETAILED ENVIRONMENT SETTING

For all three topologies, time delay is the same: 2 ticks for routers, 5 ticks for links and 10 ticks for rewards.

Link topologies of Simple Topology in Figure 1 are: $C_{ac} = 100.0$, $C_{ae} = 200.0$, $C_{be} = 200.0$, $C_{bd} = 100.0$, $C_{ec} = 100.0$, $C_{ef} = 200.0$, $C_{ed} = 100.0$, $C_{fc} = 100.0$, $C_{fd} = 100.0$.

Link topologies of Moderate Topology in Figure 2 are: $C_{ac} = 200.0$, $C_{ae} = 400.0$, $C_{be} = 400.0$, $C_{bd} = 200.0$, $C_{ec} = 200.0$, $C_{ef} = 400.0$, $C_{ed} = 200.0$, $C_{fc} = 300.0$, $C_{fd} = 300.0$, $C_{cf} = 100.0$, $C_{df} = 100.0$.

Link topologies of Complex Topology in Figure 3 are: $C_{a1} = 400.0$, $C_{a2} = 600.0$, $C_{b2} = 600.0$, $C_{b3} = 400.0$, $C_{16} = 100.0$, $C_{14} = 300.0$, $C_{24} = 600.0$, $C_{25} = 600.0$, $C_{35} = 300.0$, $C_{38} = 100.0$, $C_{46} = 300.0$, $C_{47} = 600.0$, $C_{57} = 600.0$, $C_{58} = 300.0$, $C_{6c} = 400.0$, $C_{7c} = 600.0$, $C_{7d} = 600.0$, $C_{8d} = 400.0$.

Synthetic flow demands of Simple Topology in Figure 1 are: $flow_{ac} = 90.0 + sin() * 80$, $flow_{bd} = 180.0 + cos() * 30$.

Synthetic flow demands of Moderate Topology in Figure 2 are: $flow_{ac} = 180.0 + cos() * 30$, $flow_{ad} = 90.0 + sin() * 80$, $flow_{bd} = 90.0 + sin() * 80$, $flow_{ac} = 180.0 + cos() * 30$.

Synthetic flow demands of Complex Topology in Figure 3 are: $flow_{ac} = 300.0 + cos() * 50$, $flow_{ad} = 150.0 + sin() * 140$, $flow_{bd} = 150.0 + sin() * 140$, $flow_{ac} = 300.0 + cos() * 50$.

Real flow trajectories from the American Abilene Network are shown in Figure 7. Note that we normalize the flow demands so that they can be consistent with link capacities.

To test the learned policies, we randomly change the flow demands of each IE-pair.

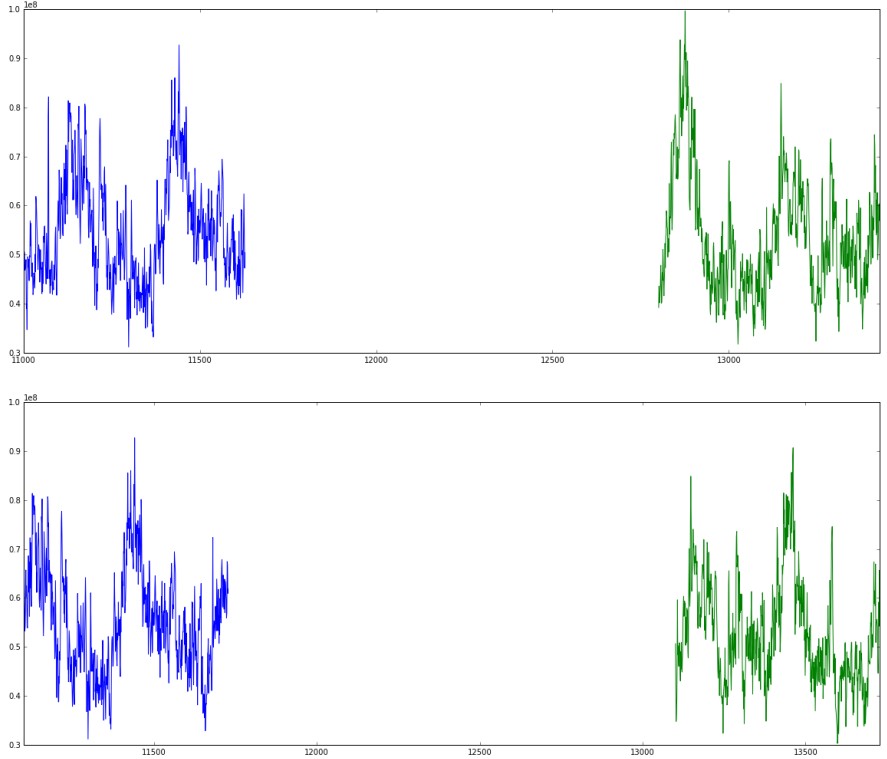

Figure 7: Real flow trajectories.

## A.2 DETAILED EXPERIMENT SETTING

State is represented as tuple <F, U, A, aveU>, where F stands for two ticks history of flow demands; U stands for five ticks history of direct link utilization ratio; A stands for last action token by the agent; aveU stands for ten ticks average of direct link utilization ratio. Specifically, for Simple Topology in Figure 1, the state dimensions of agent a, b and e are 28, 28 and 41, respectively; for Moderate Topology in Figure 2, the state dimensions of agent a, b and e are 30, 30 and 41, respectively; for Complex Topology in Figure 3, the state dimensions of agent a, b, 1, 2, 3, 4, and 5 are 30, 30, 28, 30, 30, 30 and 30, respectively.

For action, as the ingress-router should generate a splitting ratio $y_i^k$ with a constraint $\sum_k y_i^k = 1$ for current traffic demand $F_i$. So the softmax activation is chosen as the final layer of actor network. This design is natural for the continuous action with sum-to-one constraint.

Settings related with ACCNet are: buffer size is 6280; batch size is 64; learning rates of actor and critic are 0.001 and 0.01, respectively; discount factor is 0.9; weight for updating target network is 0.001.

## A.3 OTHER EXPERIMENTAL RESULTS

The link utilization ratios of AC-CNet and A-CCNet testing on Complex Topology using Abilene Network flow and $dlgMixedR$ are shown in Figure 8 and 9, respectively. A-CCNet achieves smaller fluctuation range of link utilization ratio than AC-CNet, which means that A-CCNet is better than AC-CNet to deal with this environment, as claimed in the original paper (Mao et al., 2017). Besides, similar with Figure 6, all links in Figure 8 and 9 have similar utilization ratios and the trends of the curves are consistent, which means that all the links can properly share responsibility for the flow demand according to their respective capabilities.

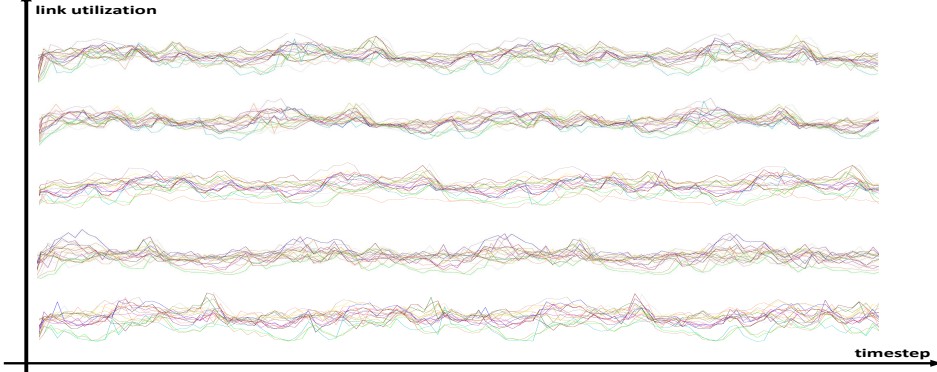

Figure 8: link utilization ratios of AC-CNet testing on real Abilene Network flow using $dlgMixedR$.

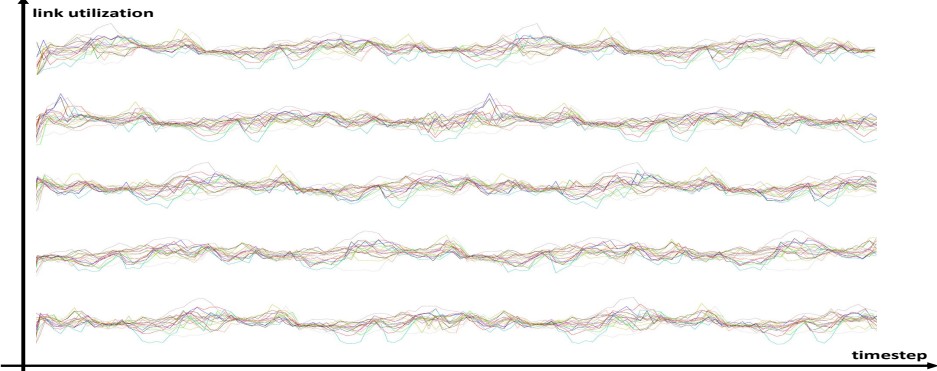

Figure 9: link utilization ratios of A-CCNet testing on real Abilene Network flow using $dlgMixedR$.

