# OpenReview forum: "Reward Design in Cooperative Multi-agent Reinforcement Learning for Packet Routing"
_ICLR.cc/2018/Conference — Reject_

### Official Review · AnonReviewer1 · 2017-11-27
**A hint a how rewards might be designed in MARL to induce cooperative behaviours, but the paper doesn't achieve as much as it could in terms of understanding and refining the general principles.**

**Rating:** 5
**Confidence:** 3

**Review:**

The authors suggest using a mixture of shared and individual rewards within a MARL environment to induce cooperation among independent agents. They show that on their specific application this can lead to a better overall global performance than purely sharing the global signal, or using just the independent rewards.

The paper is a little too focused on the packet routing example domain and fails to deliver much in terms of a general theory of reward design for cooperative behaviours beyond showing that mixed rewards can lead to improved results in their domain. They discuss what and how rewards, and this could be made more formal, as well as (at the very least) some guiding principles to follow when mixing rewards. It feels like there is a missing section between sections 2 and 3, where this methodological content could be described.

The rest of the paper has similar issues, with key intuition and concepts either missing entirely or under-represented. The technical content often assumes that the reader is familiar with certain terms, and it is difficult to see what meaningful conclusions can be drawn from the evaluation.


On a minor note, the use of the term cooperative in this paper could be better defined. In game theory, cooperative games are those in which agents share rewards. Non-cooperative (game theory) games are those where agents have general reward signals (not necessarily cooperature or adversarial). Conventionally (yes there is existing reward design/shaping literature for MARL) people have used the same terms in MARL. Perhaps the authors could define their approach as weakly cooperative, or emergent cooperation.

The related work could be better described. There are existing papers on MARL and the issues with cooperation among independent learners, and this could be referenced. This includes reward shaping and reward potential. I would also have expected to see brief mention of empowerment in this section too (the agent favouring states where it has the power to control outcomes in an information theoretic sense), as an underyling principle for intrinsic reward. However, more importantly, the authors really needed to do more to synthesize this into an overall picture of what principles are at play and what ideas/methods exist that have tried to exploit some of these principles.

Detailed comments:
  • [p2] the authors say "We set the meta reward signals as 1 - max(U l ).", before they define what U_l is.
  • [p2] we have "As many applications in the real world can be modeled using similar
methods, we expect that other fields can also benefit from this work." This statement is too vague, and the authors could do more to identify which application areas might benefit.
  • [p3, first para] "However, the reward design studies for MARL is so limited." Drop the word 'so'. Also, I would argue that there have been quite a few (non-deep) discussions about reward design in MARL, cooperative, non-cooperative and competitive domains.
  • [p3, sec 2.2] "This makes the diligent agents confuse about..." should be "confused", and I would advise against anthropomorphism at least when the meaning is obscured.
  • [p3, sec 3] "After having considered several other options, we finally choose the Packet Routing Domain as our experimental environments." Not sure what useful information is being conveyed here.
  • [sec 3] THe domain could be better described with intuition and formal descriptions, e.g. link utilization ratio, etc, before.
  • [p6] "Importantly, the proposed blR seems to have similar capacity with dlR," The discussion here is all in terms of the reward acronyms with very little call on intuition or other such assistance to the reader.
  • [p7] "We firstly try gR without any thinking" The language could be better here.

---

> ### Author Response · Authors · 2017-12-02
> **The review is very pertinent. Thanks.**
>
> The review is very pertinent. Thanks.
>
> The paper doesn't give a general principle for reward design. It only test and verify that the adaptive rewards are better (can achieve higher convergence rate and lower max link utilization ratio) than mixed rewards as well as the global and local rewards. Those rewards are tricks in some extend.
>
> The paper need further revise indeed. Thanks again for useful comments.

---

### Official Review · AnonReviewer2 · 2017-11-27
**Hard to read paper with no clear contribution**

**Rating:** 2
**Confidence:** 4

**Review:**

The paper provides an empirical study of different  reward schemes for cooperative multi-agent reinforcement learning.  A number of alternative reward schemes are proposed, partly based on existing literature. These reward schemes are evaluated empirically in a packet routing problem.

The approach taken by this paper is very ad-hoc. It is not clear to me that this papers offers any general insights or methodologies for reward design for MARL. The only conclusions that can be drawn from this paper is which reward performs best on these specific problem instances(and even this is hard to conclude from the paper).

In general, it seems strange to propose the packet routing problems as benchmark environments for reward design. From the descriptions in the paper these environments seem relatively complex and make it difficult to study the actual learning dynamics. The results shown provide global performance but do not allow to study specific properties.

The paper is also quite hard to read.  It is littered with non-intuitive abbreviations. The methods and experiments are poorly explained. It claims to study rewards for multi-agent reinforcement learning, but never properly details the learning setting that is considered or how this affects the choice of rewards.  Experiments are mostly described in terms of method A outperforms method B. No effort is made to investigate the cause of these results or to design experiments that would offer deeper insights. The graphs are not properly labelled, poorly described in general and are almost impossible to interpret. The main results are presented simply as a large table of raw performance numbers. This paper does not seem to offer any major fundamental or applied contributions.

---

> ### Author Response · Authors · 2017-12-02
> **The review is very pertinent. Thanks.**
>
> The review is very pertinent. Thanks.
>
> The paper doesn't give a general principle for reward design. It only test and verify that the adaptive rewards are better (can achieve higher convergence rate and lower max link utilization ratio) than mixed rewards as well as the global and local rewards. Those rewards are tricks in some extend.
>
> The paper need further revise indeed. Thanks again for useful comments.

---

### Official Review · AnonReviewer3 · 2017-11-27
**My review**

**Rating:** 5
**Confidence:** 2

**Review:**

The authors study the problem of distributed routing in a network, where the goal is to minimize the maximal load (i.e. the load of the link with the highest utilization). The authors advocate to use multi-agent reinforcement learning. The main idea put forward by the authors is that by designing artificial rewards (to guide the agents), one can achieve faster exploration, in order to reduce convergence time.

While the authors put forward several interesting ideas, there are some shortcomings to the present version of the paper, including:
- The design objective seems flawed from the networking point of view: while minimizing the maximal load of a link is certainly a good starting point (to avoid instable queues) one typically wants to minimize delay (or maximize flow throughput). Indeed, it is possible to have a larger maximal load while reducing delay in many cases.
- Furthermore, the authors do not provide a baseline to which the outcome of the learning algorithms they propose: for instance how does their approach compare to simple policies (those are commonplace in networking) such as MaxWeight, Backpressure and so on ?
- The authors argue that using multi-agent learning is more desirable than single agent (i.e. with a single reward signal which is common to all agents). However, is multi-agent guaranteed to converge in such a setting ? If some versions of the problem (for some particular reward signal) are not guaranteed to converge, it is difficult to understand whether "convergence"  is slow due to an inefficient exploration, or simply because convergence cannot occur in the first place.
- The learning algorithms used are not clearly explained: the authors simply state that they use "ACCNet" (from some unpublished prior work), but to readers unfamiliar with this algorithm, it is difficult to judge the contents of the paper.
- In the numerical experiments, what is the "convergence rate" ? is it the ratio between the mean reward of the learnt policy and that of the optimal ? For how many time steps are the learning algorithm run before evaluating their outcome ? What are the meaning of the various input parameter of ACCnet, and is the performance sensitive to those parameters ?

---

> ### Author Response · Authors · 2017-12-02
> **The review is very pertinent. Thanks.**
>
> The review is very pertinent. Thanks.
>
> The paper doesn't give a general principle for reward design. It only test and verify that the adaptive rewards are better (can achieve higher convergence rate and lower max link utilization ratio) than mixed rewards as well as the global and local rewards. Those rewards are tricks in some extend.
>
> The paper need further revise indeed. Thanks again for useful comments.

---

### Decision · Program_Chairs · 2018-01-29
**ICLR 2018 Conference Acceptance Decision**

**Decision:**

Reject

**Comment:**

All reviewers are unanimous that the paper is below threshold for acceptance.  The authors have not provided rebuttals, but merely perfunctory generic responses.

I think the most important criticism is that the approach is "very ad-hoc."  I would encourage the authors to consider more principled ways of automatically designing reward functions, like for example, Inverse Reinforcement Learning, in which you start with a good agent behavior policy, and then estimate a reward function for which the behavior policy maximizes the reward function.